# Evaluation of Green Super Rice Lines for Agronomic and Physiological Traits under Salinity Stress

**DOI:** 10.3390/plants11111461

**Published:** 2022-05-30

**Authors:** Muhammad Ammar Amanat, Muhammad Kashif Naeem, Hussah I. M. Algwaiz, Muhammad Uzair, Kotb A. Attia, Muneera D. F. AlKathani, Imdad Ulah Zaid, Syed Adeel Zafar, Safeena Inam, Sajid Fiaz, Muhammad Hamza Arif, Daniyal Ahmad, Nageen Zahra, Bilal Saleem, Muhammad Ramzan Khan

**Affiliations:** 1National Institute for Genomics and Advanced Biotechnology (NIGAB), National Agricultural Research Centre (NARC), Park Road, Islamabad 45500, Pakistan; mailtoammar1@gmail.com (M.A.A.); kashifuaar@gmail.com (M.K.N.); uzairbreeder@gmail.com (M.U.); imdadcas@gmail.com (I.U.Z.); syedz@ucr.edu (S.A.Z.); safeena.inam@gmail.com (S.I.); hamza.arif0305@gmail.com (M.H.A.); ad.daniyal1995@gmail.com (D.A.); nageenzahra@hotmail.com (N.Z.); bilal.saleem@bs.qau.edu.pk (B.S.); 2Department of Biology, College of Science, Princess Nourah Bint Abdulrahman University, P.O. Box 84428, Riyadh 11671, Saudi Arabia; mdfkathani@pnu.edu.sa; 3Center of Excellence in Biotechnology Research, King Saud University, P.O. Box 2455, Riyadh 11451, Saudi Arabia; kattia1.c@ksu.edu.sa; 4Rice Biotechnology Lab, Rice Department, Field Crops Research Institute, ARC, Sakha 33717, Egypt; 5Department of Plant Breeding and Genetics, The University of Haripur, Haripur 22620, Pakistan; sfiaz@uoh.edu.pk; 6Department of Biochemistry and Molecular Biology, University of Gujrat, Hafiz Hayat Campus, Jalalpur Road, Gujrat 50700, Pakistan; 7Department of Plant Breeding and Genetics, Sindh Agricultural University, Tandojam, Hyderabad 70060, Pakistan

**Keywords:** green super rice, salinity stress, salinity stress, field evaluation, gene expression

## Abstract

Rice (*Oryza sativa*) is an important staple food crop worldwide, especially in east and southeast Asia. About one-third of rice cultivated area is under saline soil, either natural saline soils or irrigation with brackish water. Salinity stress is among the devastating abiotic stresses that not only affect rice growth and crop productivity but also limit its cultivation area globally. Plants adopt multiple tolerance mechanisms at the morphological, physiological, and biochemical levels to tackle salinity stress. To identify these tolerance mechanisms, this study was carried out under both a controlled glass house as well as natural saline field conditions using 22 green super rice (GSR) lines along with two local varieties (“IRRI 6 and Kissan Basmati”). Several morpho-physiological and biochemical parameters along with stress-responsive genes were used as evaluation criteria under normal and salinity stress conditions. Correlation and Principal Component Analysis (PCA) suggested that shoot-related parameters and the salt susceptible index (SSI) can be used for the identification of salt-tolerant genotypes. Based on Agglomerative Hierarchical Cluster (AHC) analysis, two saline-tolerant (“S19 and S20”) and saline-susceptible (“S3 and S24”) lines were selected for further molecular evaluation. Quantitative RT-PCR was performed, and results showed that expression of 1-5-phosphoribosyl -5-5-phosphoribosyl amino methylidene amino imidazole-4-carboxamide isomerase, DNA repair protein recA, and peptide transporter PTR2 related genes were upregulated in salt-tolerant genotypes, suggesting their potential role in salinity tolerance. However, additional validation using reverse genetics approaches will further confirm their specific role in salt tolerance. Identified saline-tolerant lines in this study will be useful genetic resources for future salinity breeding programs.

## 1. Introduction

Salinity poses a worldwide problem to both irrigated and rainfed lands. Soil salinity affects 20% of the cultivated lands in the world leading to alkalinity and waterlogging [1]. According to the Food and Agriculture Organization of the United Nations (FAO), salinity has damaged 397 million hectares (M ha) of land area. Twenty percent of the 230 M ha of land used for crop production is affected by salinity with a staggering cost of USD 11 billion per year [2]. It is mostly caused by the natural climatic factors i.e., due to poor drainage systems and flooding with salt-rich water increased the pollution of rivers, aquifers, and seawater [3]. Sodium concentration of more than 4 dSm^−1^ or 40 mM, is primarily the leading cause of soil and water salinity [4,5].

Rice is a very sensitive crop to salinity stress [5,6,7]. A salt level of 10 dSm^−1^ causes the death of rice seedlings [8], while 3.5 dSm^−1^ salt level stress at the reproductive stage leads to a high yield loss of up to 90% [9]. Heavy concentration of sodium ions in soil lowers the capacity of plant roots to take up the water and minerals leading to retardation of plant growth and yield [5,10]. It impacts plant morpho-physiological traits i.e., root growth, plant growth, plant height, number of tillers, leaf features, panicle fertility, and seed development. Salinity causes ionic and osmotic stress in plants. To combat this, plants close their stomata for water conservation and limit transpiration, which is involved in the movement of sodium ions from roots to the shoots of plants [11]. Few rice genotypes using sophisticated physiological mechanisms including sodium (Na^+^) and potassium (K^+^) compartmentalization into apoplasts, sequestration into older tissues, upregulation of antioxidants, stomatal responsiveness and sodium exclusions have developed tolerance against salinity [12,13,14].

In plants, selective membranes for sodium ions transportation have not yet been discovered. Sodium ions are taken up into cells through nonselective cation channels i.e., Na^+^, K^+^, and Ca^+^ transporters [11,15]. Transporters are involved in salinity tolerance by reducing the build-up of Na^+^ ions through ion exclusion [16]. One of the best-known transporters involved in Na^+^/K^+^ homeostasis is high-affinity K^+^ transporters (*HKT*) including: *AtHKT1.1* [17] and *OsHKT1;5* [14]. Secondly, compartmentalization of sodium ions in vacuoles is one of the salinity tolerance mechanisms. Na^+^/H^+^ transporters (*NHX*, *OsNHX1*) present in tonoplast perform the selective sequestration of sodium ions in vacuoles [16]. In association with this process, there is an accumulation of potassium ions and other solutes in the cytosol to stabilize the osmotic pressure in the vacuoles [5]. Some other cloned genes in rice includes; *OsSKC1* [14], *OsMAPK33* [18], *OsEATB* [19], *OsCCCl^−^* [20], *OsbZIP23* [21], and *OsHAK21*/*qSE3* [22], which plays key role in salt-tolerance by increasing sodium ion uptake under salinity stress. Hence, identifying the salt-tolerant lines and evaluating the novel genetic factors involved in salt tolerance has become one of the rice molecular breeding goals.

In this regard, green super rice has been developed by pyramiding the genes that could stand with biotic and abiotic stresses and high nutrient use efficiency [23,24]. Based on yield performance, we selected 22 out of 552 green super rice (GSR) lines to evaluate their morpho-physiological and molecular response in natural saline soil and glasshouse experiments. In addition, expression patterns of novel DEGs (differential expressed genes) were assessed among the salinity-tolerant and -sensitive lines to understand the genetic mechanism underlying these novel genes in response to salinity tolerance. This study identified the salinity tolerant GSR lines based on glasshouse and field experiments and provided the knowledge about key morpho-physiological traits which can be used in the future for salinity experiments and help to understand the genetic role of novel salinity tolerance related genes.

## 2. Materials and Methods

### 2.1. Plant Material and Experiments

Twenty-two GSR lines reported previously [25] along with two local varieties (“IRRI 6 and Kissan Basmati”) were selected and screened for salinity tolerance under different conditions (Appendix A). For this purpose, three experiments i.e., glasshouse screening, field evaluation, and molecular evaluation were performed to identify the salt-tolerant and salt-susceptible GSR lines and understand the genetic mechanism underlying the salt tolerance in rice.

### 2.2. Glasshouse Experiment

#### Seedling Evaluation at 200 mM NaCl

Seeds of 22 GSR lines along with two local varieties (“IRRI 6 and Kissan Basmati”) were surface sterilized with 30% hydrogen peroxide solution for 10 min followed by thorough washing with autoclaved distilled water. After sterilization, seeds were placed in Petri dishes for 72 h in a growth chamber at 24–28 °C with 14 h light period and 10 h dark period. Three days old uniform seedlings were transferred to plastic tubs containing 100 L nutrition medium (Yoshida medium). Plastic tubs were kept in a glasshouse under controlled conditions with a light period of 14 h followed by a 10 h dark period, humidity 55% to 65%, and temperature 26–28 °C.

Previously, 200 mM salt concentration is reported for screening of rice genotypes [26]. Rice seedlings were normally grown for 14 days. After that one batch of eight seedlings per genotype was left for normal growth (control), while the second batch consisting of eight seedlings per genotype was treated with 200 mM salt concentration (commercial grade NaCl) for 7 days, to evaluate the genotypes rigorously. For this purpose, commercial-grade salt (solid) was dissolved in 2 liters of water to make the salt solution and this was applied to the plants. A randomized complete block design with three replicates was followed. After 7 days of salt stress, seedlings were evaluated for morpho-physiological traits i.e., shoot length (SL, cm), root length (RL, cm), total plant length (TPL, cm), shoot fresh weight (SFW, g), shoot dry weight (SDW, g), root fresh weight (RFW, g), and root dry weight (RDW, g). For this purpose, five seedlings from each replicate were selected randomly and lengths were measured with the help of a graduated ruler in centimeters. Similarly, weights were also recorded with electronics balance (Compax, RS 232C). For dry weights measurements, 3 days oven-dried (37 °C) seedlings were used. Rice seedlings were screened (according to SES, IRRI) for the salt toxicity symptoms. Three replicates of root and shoot-related tissues from each treatment were analyzed for sodium (Na^+^) and potassium (K^+^) concentration. The root and shoot tissues were rinsed with distilled water several times. The samples were kept in an oven for 72 h at 60 °C. After the samples had dried, the sodium and potassium concentrations (SNC, SKC, RNC, and RKC) were measured according to the already proposed method by using a flame photometer [27]. The concentration of Na^+^ and K^+^ in root and shoot tissues were summed to make the whole plant Na^+^ and K^+^. The whole plant Na^+^/K^+^ ratio (NKR) was also calculated.

### 2.3. Field Experiment

#### 2.3.1. Plant Growth and Field Conditions

The healthy seeds of 22 GSR lines and two local varieties (“IRRI 6 and Kissan Basmati”) were sown in nursery trays during the month of June 2021. Nursery trays were filled with a mixture of soil, peat moss, and sand in a ratio of 2:2:1. At the three-leaf seedling stage about 30 days after sowing (DAS), transplantation to the control field and natural saline conditions (Appendix A) were carried out at the Soil Salinity Research Institute, Pindi Bhattian (31.472° N, 74.243° E), Pakistan. The experiment was designed according to a split-plot randomized complete block design (RCBD) with three replicates for each treatment.

#### 2.3.2. Field Evaluation for Morphological Traits

At the maturity stage, plant height (PH), and the number of fertile tillers (TN) from five randomly selected plants of each replication were recorded for studied genotypes. After harvesting, grain yield per plant (GY, g), straw yield per plant (SY, g), seed length (GL, mm), 1000 grain weight (TGW, g), harvest index (HI, %), and stress susceptibility index (SSI) were recorded. GL was measured with the help of a digital vernier caliper in millimeters. At the same time, the GY, SY, and TGW were recorded with electronics balance (Compax, RS 232C). SSI for grain yield was calculated by using the following equation.
(1)$$\mathrm{SSI}=\frac{1-\frac{Ys}{Yp}}{D}$$
*Y_s_* and *Y_p_* is the yield under stress and control field conditions, respectively, while *D* is a stress intensity $\left(1-\frac{X}{Xp}\right)$ as described previously [25].

#### 2.3.3. Selection Criteria of Tolerant and Susceptible Lines

Morpho-physiological notes from both the glasshouse experiment (SL, RL, TPL, SFW, RFW, SDW, RDW, SNC, SKC, RNC, RKC, and NKR) and the field trial (PH, TN, GY, SY, TGW, GL, and HI) was employed to evaluate the performance of 22 GSR lines and two local varieties under salt stress treatments. Agglomerative Hierarchical Cluster (AHC) analysis in XLSTAT was used for the characterization of rice genotypes [28]. For dissimilarities calculation, the Euclidean distances were measured by applying Ward’s method [29].

### 2.4. Molecular Characterization

#### 2.4.1. RNA Sampling, Extraction, cDNA Synthesis

The roots of the seedlings were thoroughly washed with distilled water. After root disinfection, the root tissues from the selected salt-tolerant (“S19 and S20”) and salt-susceptible lines (“S3 and S24”) were immediately stored in liquid nitrogen. A sampling of lateral root tissues was carried out from control and salt stress plants with three biological replicates. The total RNA was extracted using the TRIzol method devised by [30]. RNA quantification was performed using the nano drop (Biospec™ nano spectrophotometer for life sciences) for sample normalization. Complementary DNA was synthesized as described by [31]. In detail, Thermo Scientific Revert Aid reverse transcriptase-III, First Strand cDNA Synthesis Kit (K1691) was used to synthesize cDNA from the extracted RNA.

#### 2.4.2. Selection of Differentially Expressed Genes (DEGs) and Expression Analysis

The transcriptome data from two salt studies (EMBL ArrayExpress E-MTAB-10653 https://www.ebi.ac.uk/arrayexpress/experiments/E-MTAB-10653/ (accessed on 1 March 2022) and NCBI GEO GSE60287 https://www.ncbi.nlm.nih.gov/geo/query/acc.cgi?acc=GSE60287 (accessed on 1 March 2022) was acquired [32,33]. Five differentially expressed genes (DEGs) (*LOC_Os05g33260*, *LOC_Os03g43850*, *LOC_Os05g04830*, *LOC_Os01g04950 (PTR2)*, and *LOC_Os08g34540*) with higher log 2 foldchange values were selected, to understand their role in salt tolerance. Gene-specific pairs of primers were designed using freely available AmplifX (v.1.7.0) online software (https://amplifx.software.informer.com/1.7/, accessed on 1 March 2022, Appendix A). The specificity of the designed primers was tested by UCS-PCR at UCSC-In Silico PCR genome browser (https://genome.ucsc.edu/ (accessed on 1 March 2022)) [34]. To our knowledge, these selected genes have not been evaluated before for salt tolerance. According to the rice genome annotation project these selected genes are involved in different molecular functions, biological processes, cellular components, and responses to different stresses through transportation. In addition, we studied the already characterized three salt tolerance-related genes *OsSKC1* [14], *OsHAK21*/*qSE3* [22], and *OsbZIP23* [21] played a significant role in salt stress tolerance in rice. Quantitative real time-PCR (qRT-PCR) was performed to determine the relative expression pattern of selected genes on StepOne RT-PCR (Applied Biosystems^®^ 7900 HT Fast RT-PCR). *OsActin1* was used as an internal reference gene for data normalization [35]. The 2−^ΔΔ^CT method was employed to estimate the relative expression pattern.

### 2.5. Statistical Analysis

Analysis of variance (ANOVA) was employed to estimate the significant variance of the traits and genotypes. Violin plot and Correlation among the traits were calculated using “ggcorrplot2” (https://github.com/caijun/ggcorrplot2, accessed on 1 April 2022) function in the R software (https://cran.r-project.org/bin/windows/base/, accessed on 1 April 2022). Heritability was also calculated by using the “heritability” package of R [36,37]. XLSTAT was used for principal component analysis (PCA), to classify the various morpho-physiological traits and genotypes [38]. For comparison, analysis *t*-test was used.

## 3. Results

### 3.1. Glasshouse Experiment

#### 3.1.1. Response of Seedlings Growth under Glasshouse Conditions

Twenty-four GSR lines including two local varieties were screened in a greenhouse. Collected data were subjected to analysis of variance (ANOVA) and mean squares for shoot length (SL), root length (RL), total plant length (TPL), shoot fresh weight (SFW), root fresh weight (RFW), shoot dry weight (SDW), root dry weight (RDW), shoot sodium concentration (SNC), shoot potassium concentration (SKC), root sodium concentration (RNC), root potassium concentration (RKC), and whole plant sodium/potassium ratio (NKR) are presented in Table 1. Results revealed that genotypes (G) varied highly significantly (*p* < 0.001) for all the parameters except for RL and RDW (Table 1). Similarly, the treatments (T) were also highly significantly (*p* < 0.001) varied for all the parameters except for RDW. We also studied the genotypes × treatments interactions. Results indicate that SDW, SNC, SKC, RNC, RKC, and NKR showed highly significant differences (*p* < 0.001), SL demonstrated significant differences (*p* < 0.05) for G × T, while the rest of the parameters were non-significant (Table 1).

#### 3.1.2. Mean Variability of Seedlings Parameters

Under control conditions, the SL changes significantly and showed a 5.65% reduction in saline samples as compared to the control condition (Appendix A and Figure 1). The genotypes “S13” (29.91 cm) and “S1” (28.52 cm) showed higher SL (Table 2). RL varied under both conditions. Overall, RL showed an 11.20% reduction as compared to control (Appendix A). Under saline conditions, the genotypes “S5” (21.16 cm) and “S3” (20.83 cm) were at maximum RL (Table 2). Both SL and RL combine and make TPL. The overall mean was 8.10% reduced in saline conditions (Appendix A and Figure 1). Under saline conditions, the genotypes “S4” (49.35 cm) and “S5” (48.33 cm) were at maximum TPL (Table 2). The mean of SFW and RFW in saline conditions decreased up to 38.39% and 45.66%, respectively (Appendix A and Figure 1). The genotypes “S4” and “S5” followed by “S3” showed maximum SFW (Table 2). Similarly, the mean of RDW under saline was reduced by 22.63% as compared to the control (Appendix A and Figure 1). The lines “S18” (2.23), “S10” (0.022), and “S24” (0.020) were the outperformers for RDW (Table 2).

Sodium (Na^+^) and potassium (K^+^) are very important parameters for the identification of salt-tolerant lines. Under control and saline conditions, SNC and RNC change significantly. In saline conditions, the means of SNC and RNC were increased up to 219.42% and 102%, respectively. In saline conditions, the genotypes “S17” (1.70), “S22” (1.67), and “S14” (1.62) absorbed more sodium (Na) in their roots (Table 2). Similarly, we also checked potassium (K) in the root samples. The overall mean of RKC was increased by 34.10% in saline conditions as compared to the control mean, while the mean of SKC was reduced by 30.54% as compared to the control (Appendix A and Figure 1). The mean of plants in saline conditions showed a 270% increase in NKR as compared to plants in control conditions (Appendix A and Figure 1). The genotypes “S15” (2.34), “S10” (2.15), and “S2” (2.12) absorbed more Na^+^ and less K^+^ as compared to control plants. These genotypes were declared as salt-tolerant genotypes for NKR. On the other hand, the genotypes “S3” (0.76), “S16” (0.83), and “S4” (1.01) were in the opposite direction for performance (Table 2).

#### 3.1.3. Heritability (H^2^) of Seedlings Parameters

Heritability is believed to be a fundamental sign for the development of a further developed populace. Determination of a single plant at the seedling stage can be demonstrated as more functional for a quality that is profoundly heritable when contrasted with less heritable characteristics. Heritability calculates under 200 mM salt stress were > 50% (Table 1), for example: SL (72.68%), SDW (72.39%), SKC (71.01%), SFW (65.60%), RFW (52.69%), and followed by TPL (50.09%). This indicates that over 50% of hereditary changes transmitted to offspring were additive in nature. In this way, selection for these parameters might be perceived as valuable during early generations.

#### 3.1.4. Correlation among Seedlings Parameters

It is reported that tolerant genotypes showed longer roots and higher weights. To study the relative importance of the different traits, a correlation was carried out. Pearson’s correlation results implied that RL showed a very low correlation with all the studied parameters except TPL, SFW, RFW, and SDW (Figure 2). RL is a very important trait because it helps plants to absorb nutrients. Reduction in RL and root weights due to no production of new roots indicates that this is a salt-sensitive trait, so in an assortment of salt-tolerant genotypes, these traits cannot be used. SL showed highly significant (*p*-value < 0.001) correlation with TPL (*r* = 0.86) and SDW (*r* = 0.71), significant (*p*-value < 0.01) with SFW (*r* = 0.58) and RFW (*r* = 0.51), and significant (*p*-value < 0.05) with RNC (*r* = 0.40). TPL also showed highly significant (*p*-value < 0.001) correlation with SDW (*r* = 0.74), SFW (*r* = 0.71), and RFW (*r* = 0.63). On the bases of these findings SL-related traits can be used as a selection criterion.

#### 3.1.5. Principal Component Analysis of Seedlings Parameters

To evaluate the range of genotypes and association between rice seedlings parameters under control and 200 mM salt stress conditions, PCA based on the correlation matrix was used to study the variation pattern in 24 GSR lines. In this studied panel of genotypes, under control conditions first, two PCs covered 52.1% of the total variation (Figure 3A). In the control, environment PC1 accounted for 32.8% of the variance while the PC2 accounted for 19.3%. Under 200 mM salt stress conditions first two PCs covered 58.3% of the total variation (Figure 3B). PC1 and PC2 accounted for 39.8% and 18.5% of the variance under 200 mM salt stress conditions. Prognostication of characteristics on PC1 and PC2 in control and 200 mM salt stress conditions showed that SL, RL, RFW, SFW, SDW, and RFW were amazingly and emphatically connected with TPL. These outcomes are further affirmation of the progressive bunch investigation (Figure 2). The genotypes “S13”, “S10”, “S3”, “S22”, “S20”, and “S1” were inverse to genotypes “S24”, “S6”, “S18”, and “S12” in control conditions. A clear difference was present between salt-tolerant and salt-susceptible genotypes. The genotypes “S1”, “S19”, “S20”, “S23”, and “S13” were inverse to “S24”, and “S3” under the 200 mM salt stress conditions.

### 3.2. Field Experiment

#### 3.2.1. Effects of Salinity on Yield Components under Field Conditions

Highly significant (*p* < 0.001) differences in plant height (PH), tillers number (TN), grain yield (GY), straw yield (SY), 1000-grain weight (TGW), grain length (GL), and harvest index (HI) were observed between genotypes evaluated in this study (Table 3). Between control and salinity stress, all the studied parameters differ highly significantly (*p* < 0.001) except PH and GL (Table 3). The interaction between genotypes and treatment was non-significant for all the parameters except for HI (Table 3).

There was a wide variation among the genotypes. PH was influenced by salinity slightly and reduced by 0.92% (Appendix A) under 200 mM salt stress conditions. The highest PH was recorded in genotypes “S3” (117.53), “S23” (114), and “S24” (113.8) (Table 4). Tiller numbers were reduced by 27.02% in salt conditions (Figure 4 and Appendix A). The genotypes “S9” (22.2), “S19” (22), and “S24” (21.8) performed better (Table 4). Results from Table 3 showed that the GY was influenced by 200 mM salt stress. The mean of GY under 200 mM salt stress was reduced by 59.16% as compared to the mean of control (Appendix A). Table 4 showed that the genotypes “S9” (51.46), “S8” (48.8), and “S22” (42.66) performed well under 200 mM salt stress and have higher GY.

Under salt stress conditions, SY reduced up to 17.62% as compared to control conditions (Appendix A). Grain yield was significantly reduced under 200 mM salt stress. The mean of TGW was reduced by 10.45% under salinity (Table 4). Grain length, (GL) a component of the physical quality of rice, was not affected by the salinity. The overall mean of GL in salinity was slightly increased by 0.72% (Figure 4 and Appendix A). The genotypes “S1” (12.18), “S12” (11.14), and “S15” (10.05) showed longer grains as compared to the genotypes “S2” (7.81), “S10” (8.49), and “S23” (8.81) (Table 4). HI was influenced by salinity and its means were reduced by 33.57% as compared to the control’s mean (Figure 4 and Appendix A). Under salinity, the genotypes “S12” (0.41) and “S14” (0.37) performed well (Table 4). SSI was also calculated and on the bases of SSI the genotypes “S19” (1.16), “S18” (0.94), and “S23” (0.94) were recognized as salt-tolerant (Table 4).

#### 3.2.2. Association among Yield Parameters under Field Conditions

Association between yield related parameters such as PH, TN, GN, SY, TGW, GL, HI, and SSI were recorded under field conditions (Figure 5). Results showed that GL showed highly significant (*p* < 0.001) correlation with PH (*r* = 0.65 ***) and TGW (*r* = 0.72 ***) under salinity and control conditions. SY showed highly significant (*p* < 0.001) correlation with GY (*r* = 0.87 ***) under control conditions. Under stress conditions, the sensitive genotypes showed less SY, HI, and GY. In this study, HI also showed highly significant (*p* < 0.001) correlation with GY (*r* = 0.56 **) and SY (*r* = −0.66 ***) under salinity and significant (*p* < 0.05) with TN (*r* = 0.44 *) under control conditions (Figure 5). HI, SY, and GY can be used as a selection criterion.

#### 3.2.3. Assessment of Rice Genotypes Using Principal Component Analysis (PCA)

Principal Component Analysis (PCA) was performed to distinguish the main components of yield-related parameters of rice genotypes that best depict the reaction to salt pressure to recognize salt-resistant genotypes. Scree plots showed the variation percentage in relation to each of the PCs. The initial two PCs represented 44.9% and 22.9% of the total variance variety among rice genotypes (Figure 6A), individually all parameters were plotted against the *x*-axis and *y*-axis. PC1 represented higher values for TGW, GL, and PH which showed association among each other, while PC2 represented higher values for SY, HI, GY, and TN under control conditions (Figure 6B).

Under saline conditions, scree plot (Figure 6C) and parameters (Figure 6D) were shown. The first principal component (PC1) contributed to maximum variability (34.2%). Parameters such as PH, TGW, and GL were positively loaded. The PC2 showed 21.5% of the total variance. In PC2 the parameters HI, GY, and SSI were positively loaded. Results showed that the genotypes are more diverse from each other. Environment vacillations represent a significant danger to worldwide yield production, and genotypes might have the option to endure the unfavorable conditions coming about because of environmental change and subsequently, be valuable as possible guardians in rearing projects pointed toward improving the salinity tolerance in rice.

#### 3.2.4. Cluster Analysis of Rice Genotypes

Hierarchical clustering under saline conditions of the glasshouse as well as the field was performed. On the bases of the performance of the rice genotypes under salinity stress, 24 genotypes were divided into four groups susceptible, moderate-susceptible, moderate-tolerant, and tolerant (Figure 7). Out of 24 rice genotypes, only two genotypes fall in the first group (susceptible); 11 genotypes in the second group (moderate-susceptible); four genotypes in the third (moderate-tolerant) group; and the fourth (tolerant) group contains seven rice genotypes.

### 3.3. Molecular Characterization

#### Expression Profiling Analysis in Salt-tolerant and Susceptible Genotypes

We analyzed the physiological behavior of rice seedlings under glasshouses as well as in the field. On the bases of these findings, we characterized the rice genotypes as salt-tolerant and salt-susceptible (Figure 8A–C). To understand the mechanism of salinity tolerance in rice, five genes from the previously published RNA-seq data were selected, and previously well know salt-tolerant genes *LOC_Os01g20160* (*OsSKC1/OsHKT8)*, *LOC_Os03g37930* (*OsHAK21*/*qSE3*), and *LOC_Os02g52780* (*OsbZIP23*) also selected. Their expression was checked in salt-tolerant and salt-susceptible genotypes (Figure 8D). The genes *LOC_Os05g33260*, *LOC_Os03g43850*, *LOC_Os05g04830*, and *LOC_Os01g04950* were significantly up-regulated in salt-tolerant genotypes, while the gene *LOC_Os08g34540* was unchanged in the tolerant genotypes.

## 4. Discussion

Crops are frequently faced with different biotic and abiotic stresses that limit their development, causing impressive decreases in farming production all over the world [25,39,40]. Plants are immobile and take essential nutrients, and water from the soil. In most of the previous studies, salinity effects were studied at the seedling stage [11,41,42,43] and few studies are reported under field conditions [44,45]. During the life cycle of plants, they face different kinds of environmental stresses at every stage of growth, so it is very important to understand each developmental stage. Salinity is the most drastic abiotic stress influencing plant growth and development [46]. To balance the issues presented by expanding salinization of arable lands, there is a need to painstakingly consolidate salt-tolerant genes into cultivated genotypes and new genotypes being delivered. In accordance with this, it is vital to distinguish salt-tolerant germplasm phenotypically.

In this study, we screened rice genotypes at the seedling stage using glasshouse conditions as well as in the field conditions. Rice is very sensitive to salt stress, however, the sensitivity of rice to salt is subject to cultivar. There is a great amount of genetic diversity amongst the different crops for salt tolerance. Analysis of variance (ANOVA) is the best marker to check the response of crop plants to environmental stresses. Genotypes, treatments, and their interactions are the components of stress assessment [47,48]. Present findings demonstrated that genotype diversity significantly affected all parameters. The genotype impact is at first evaluated to decide the viability of choice. Population variation is the principal component in genotype determination [49]. The presence of variations among the genotypes and the choice of screening methods are very important. The significant impact of treatments on a parameter demonstrates that the choice of treatment is effective to study the behavior of a genotype to a parameter [47,50]. The salinity resistance at the seedling stage does not associate with resilience at other vegetative and reproductive stages in rice [51]; nonetheless, it can reduce the crop yield by adversely influencing yield-related parameters such as tiller number, plant biomass, time of flowering, and harvest index [52,53,54,55]. So, this study was conducted at seedling as well as maturity stages, and the molecular mechanism of salinity was also studied.

The rice plants under saline conditions showed significant inhibitory effects for all the growth-related parameters of the seedling as well as the maturity stage. Salinity negatively affects the shoot, root, and biomass which leads to the overall reduction in growth and production [42]. The decrease in development might be brought about by the overabundance of harmful NaCl amassing in the soil around the roots causing imbalanced supplement take-up by the seedlings [56]. A decrease in leaf area is related to changes in the leaf life cycle because of salt stress, bringing about a diminished pace of net photosynthesis [5,57]. This might be because of stomatal closer, an interior decrease of CO_2_, and diminished movement of the protein RuBisCo [58]. Upkeep of photosynthesis is vital to keep up with normal transpiration under salinity and is a significant mark of salt resistance [59]. In this study, salt-tolerant genotypes showed higher biomass as compared to salt-susceptible genotypes. Under salt stress Na^+^/K^+^ ratio increased significantly and after recovery minimum Na^+^/K^+^ ratio is a good parameter for the identification of tolerant genotypes [60,61]. In this study, under the salinity Na^+^ concentration significantly increased in shoot and roots of all the genotypes as compared to control. The ratio of Na^+^/K^+^ was also increased under salinity stress. These findings were in agreement with the previous research [5,61,62]. The present study further explained that shoot-related parameters such as SL, SFW, SDW, and SKC can be used for the selection of tolerant genotypes under glasshouse conditions.

Correlation examination can be utilized to recognize the best parameters. In this study, the investigations were focused on yield as the main parameter. Results of glasshouse evaluation showed that RL showed a very low correlation, so in an assortment of salt-tolerant genotypes these traits cannot be used. SL showed a highly significant correlation with TPL, SDW, SFW, RFW, and RNC. TPL also showed a highly significant correlation with SDW, SFW, and RFW. Association between yield-related parameters under field conditions showed that GL had a highly significant correlation with PH and TGW under salinity and control conditions. Similarly, SY showed a highly significant correlation with GY under control conditions. HI also showed a highly significant correlation with GY and SY under salinity and significant with TN under control conditions. These findings provide a basis for in-depth analysis and are also in agreement with the previous studies [63,64].

In different agricultural crops such as *Brassica napus* L. [65], *Zea mays* L. [66], *Glycine max* [67], *Triticum aestivum* [10], and *Oryza sativa* [29,68], principal component analysis (PCA) has been utilized to identify salt-tolerant lines. In this study, PCA results indicate that shoot-related parameters cluster together showing a strong correlation among them. In the field evaluation, the salt susceptible index (SSI) for each genotype was calculated for grain yield. The low values of SSI for genotypes performed better under salt stress. The genotypes “S16”, “S17”, “S18”, “S19”, “S20”, and “S6” were salt-tolerant on the basis of SSI. SSI was also used previously to calculate the stress susceptible index [69,70]. Previously it is reported that S3 is a drought-sensitive genotype [25]. In this study, we also found that S3 and S24 are salt-susceptible genotypes.

Many transcriptomic studies have been carried out to address the impact of salinity on rice. All of those studies differ from each other i.e., with the different mediums being used for growth, different plant stage, and different stress exposure time. Selected genes were novel, and their expression showed up-regulation in salt-tolerant genotypes. The most important adaptation for the plant to combat salinity is to block the transport and accumulation of Na^+^ ions in the leaves [5]. Expression of *OsHKT1;5* is correlated to the salt tolerance of a rice genotype. *OsSKC1 (OsHKT1;5)* is expressed in plant roots, where it helps the retrieval of Na^+^ ions into xylem parenchyma cells from the xylem prior to its transport in shoots [5,14]. Our results are also consistent with [71] as an increased expression is observed for the salt-tolerant lines. These findings indicate that these genes have a role in the adaptation of rice genotypes to harsh environments.

## 5. Conclusions

We tested the rice genotypes under salt stress in glasshouse and field conditions. In the glasshouse evaluation, the shoot-related parameters, while in field conditions, the grain yield and salt susceptible index can be used as a selection criterion. We additionally presume that the two investigation strategies (glasshouse and field) are similarly solid and can be utilized for tests yet work better together to affirm the exactness of test results. Based on current findings, the genotypes “S19” and “S20” were recognized as salt-tolerant while the genotypes “S3” and “S24” were salt-susceptible. Furthermore, the genes *LOC_Os05g33260*, *LOC_Os03g43850*, *LOC_Os05g04830*, and *LOC_Os01g04950* have a significant role in salt tolerance. Information from this study can assist rice growers and different researchers as they screen and select salt-tolerant rice genotypes for varietal improvement. Further examinations will be required for the functional confirmation of these genes.

## Figures and Tables

**Figure 1 plants-11-01461-f001:**
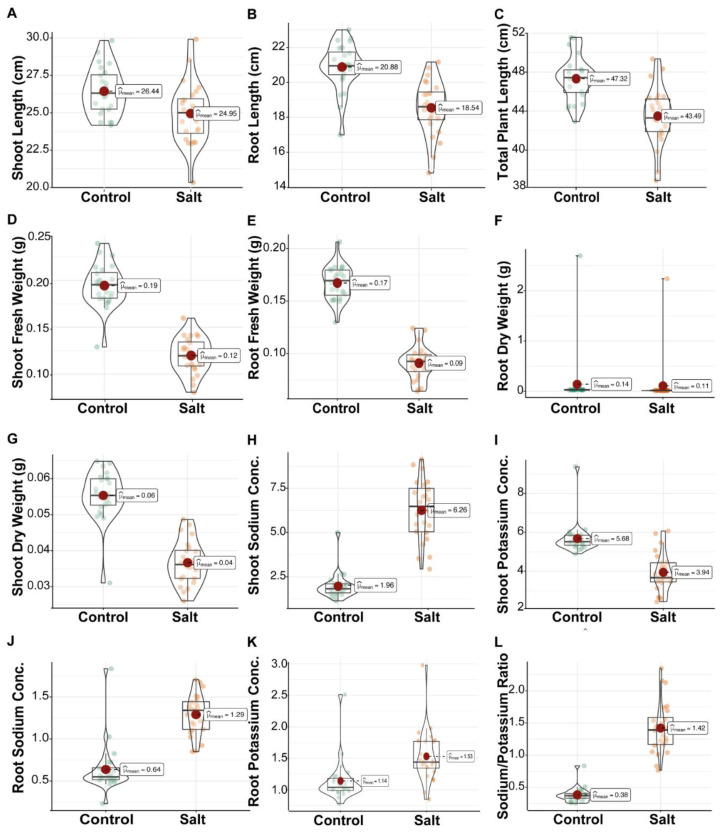
Comparison of 12 morpho-physiological traits (**A**–**L**) of rice genotypes at seedlings stage. GSR lines were evaluated in control and 200 mM salt stress (Salt) under glasshouse conditions. Circles with different colors represent the genotypes. Inside the violin plot, big red circle showed the mean, boxplot represent the median (middle line), minimum, maximum, and error bars.

**Figure 2 plants-11-01461-f002:**
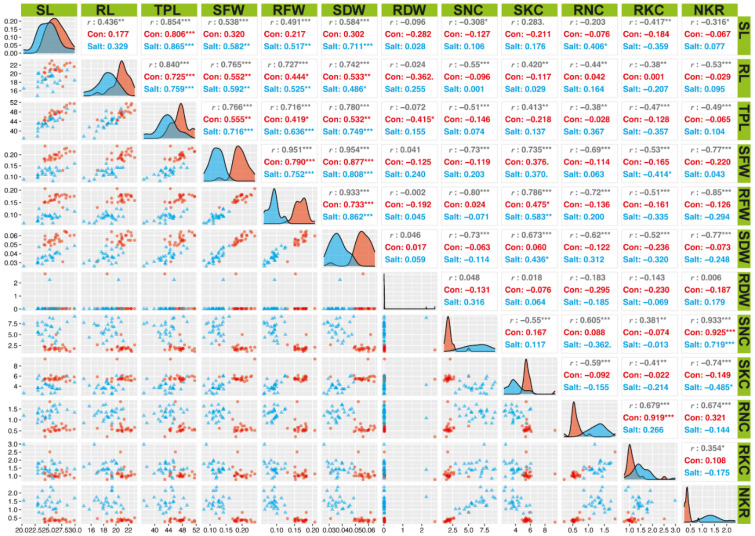
Correlation between 12 morpho-physiological traits of rice genotypes at seedlings stage. Plants were evaluated in control (Con) and 200 mM salt stress (Salt). SL = shoot length (cm); RL = root length (cm); TPL = total plant length (cm); SFW = shoot fresh weight (g); RFW = root fresh weight (g); SDW = shoot dry weight (g); RDW = root dry weight (g); SNC = shoot sodium (Na^+^) concentration; SKC = shoot potassium (K^+^) concentration; RNC = root Na^+^ concentration; RKC = root K^+^ concentration; NKR = whole plant sodium/potassium (Na^+^/K^+^) ratio. *** Highly significant (*p* < 0.001); ** Highly significant (*p* < 0.01); * significant (*p* < 0.05).

**Figure 3 plants-11-01461-f003:**
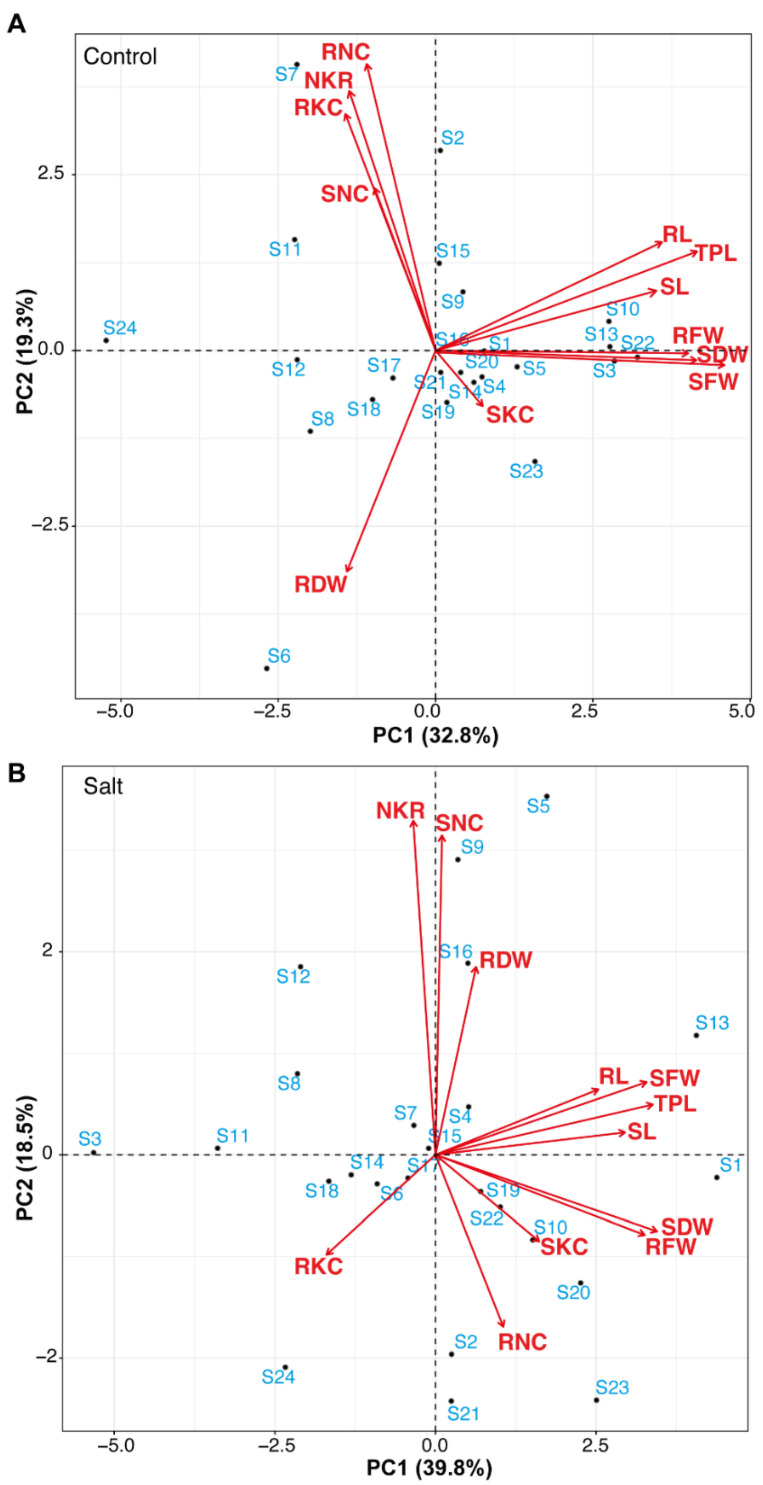
PCA of morpho-physiological traits of rice genotypes. Rice seedlings were evaluated under control (**A**) and 200 mM salt stress (**B**). SL = shoot length (cm); RL = root length (cm); TPL = total plant length (cm); SFW = shoot fresh weight (g); RFW = root fresh weight (g); SDW = shoot dry weight (g); RDW = root dry weight (g); SNC = shoot sodium (Na^+^) concentration; SKC = shoot potassium (K^+^) concentration; RNC = root Na^+^ concentration; RKC = root K^+^ concentration; NKR = whole plant sodium/potassium (Na^+^/K^+^) ratio.

**Figure 4 plants-11-01461-f004:**
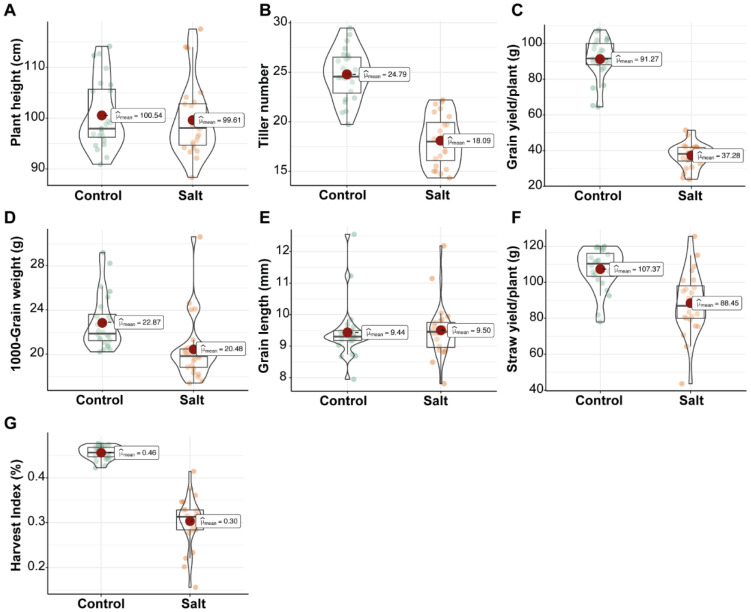
Comparison of yield related parameters (**A**–**G**) of rice genotypes at maturity stage. Plants were evaluated in control and 200 mM salt stress (salt) under field conditions. Circles with different colors represent the distribution of genotypes. Inside the violin plot, big red circle showed the mean, boxplot represent the median (middle line), minimum, maximum, and error bars.

**Figure 5 plants-11-01461-f005:**
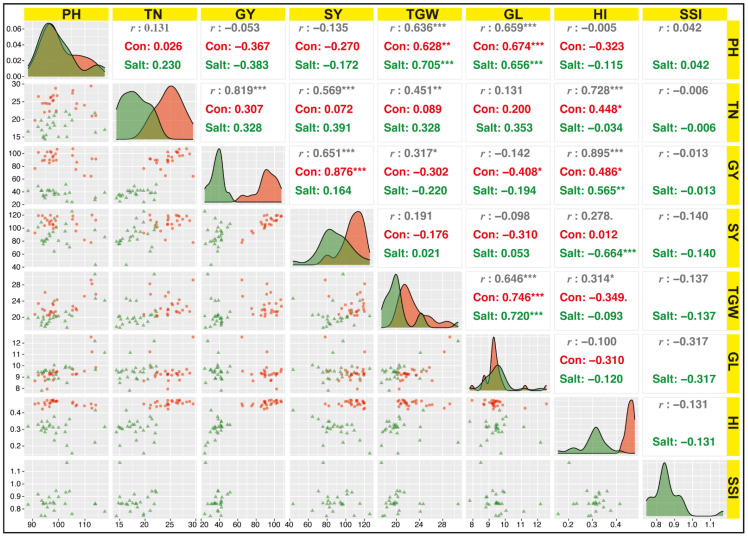
Correlation between seven morpho-physiological traits of rice genotypes at maturity stage. Plants were evaluated in control (Con) and salt stress (Salt) under field conditions. PH = plant height (cm); TN = tillers number; GY = grain yield/plant (g); SY = straw yield (g); TGW = 1000-grain weight (g); GL = grain length (mm); HI = harvest index (%). *** Highly significant (*p* < 0.001); ** Highly significant (*p* < 0.01); * significant (*p* < 0.05).

**Figure 6 plants-11-01461-f006:**
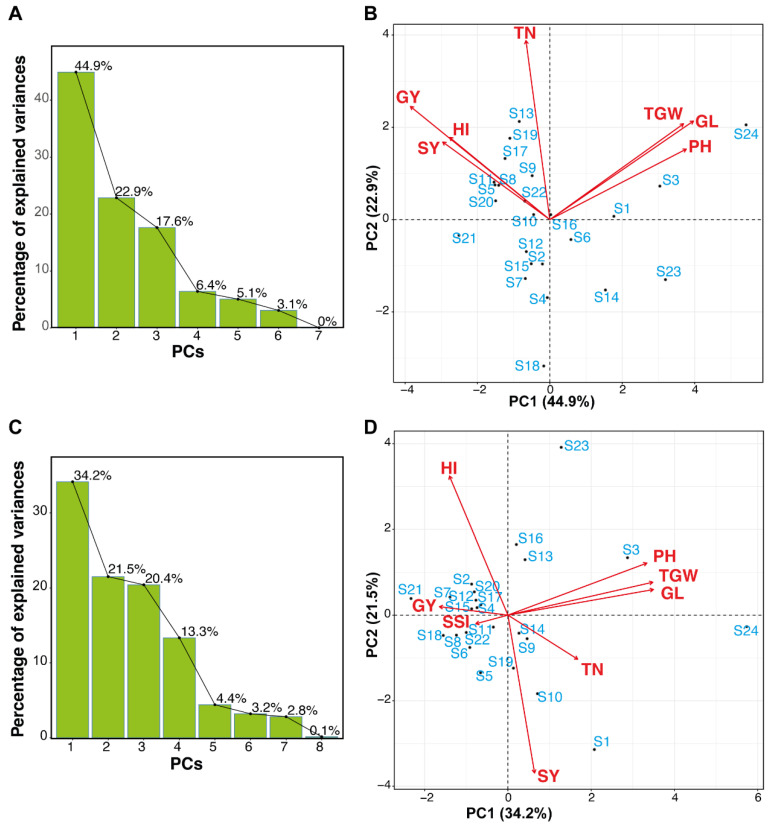
PCA of yield and yield-related traits of rice genotypes. Rice genotypes were evaluated at control (**A**,**B**) and salinity stress (**C**,**D**) under field conditions. Scree plots of control (**A**), salinity (**C**) showing significant PCs. First two PCs of control (**B**) and salt stress (**D**). PH = plant height (cm); TN = tillers number; GY = grain yield/plant (g); SY = straw yield (g); TGW = 1000-grain weight (g); GL = grain length (mm); HI = harvest index (%).

**Figure 7 plants-11-01461-f007:**
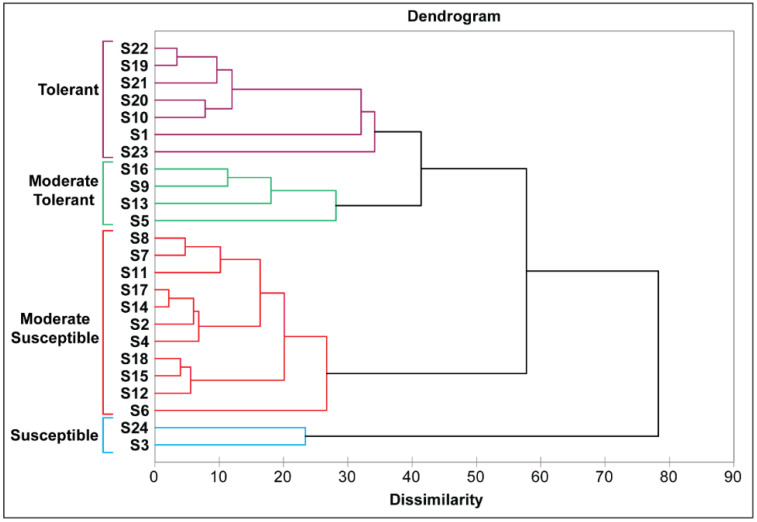
Grouping of 24 rice genotypes on the bases of their performance under salt stress conditions in glasshouse and field conditions. Four main groups light blue (susceptible); red (moderate-susceptible); light green (moderate-tolerant); and purple (tolerant) were created.

**Figure 8 plants-11-01461-f008:**
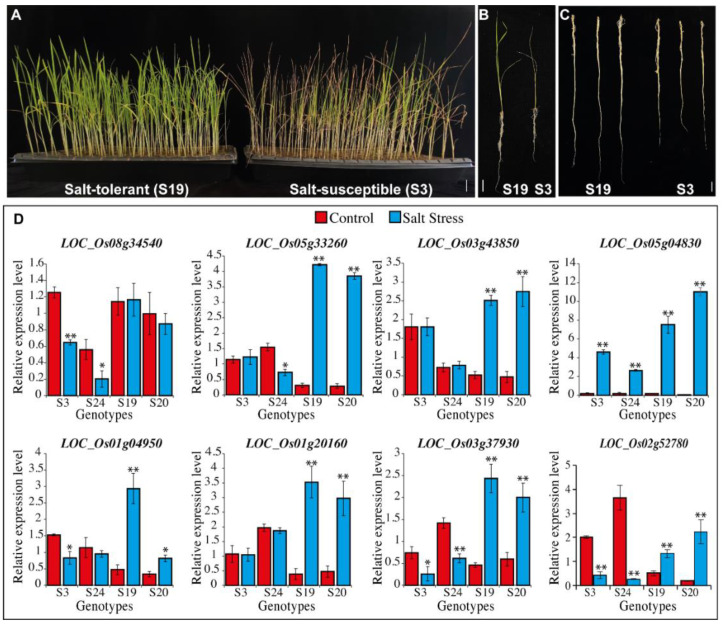
Physiological response of rice tolerant (“S19”) and susceptible (“S3”) genotypes in response to 200 mM salt stress. Comparison of S19 and S3 under salt stress (**A**), whole seedlings (**B**); and roots (**C**). Bars = 1 cm. (**D**) Quantification RT-PCR analysis of selected genes expression in salt-tolerant (“S19” and “S20”) and salt-susceptible (“S3” and “S24”) genotypes. *OsActin1* was used for normalization. Bars showing the mean ± SD of three different experiments. *t*-test was used; ** *p* < 0.01, * *p* < 0.05.

**Table 1 plants-11-01461-t001:** Mean squares values from analysis of variance (ANOVA) for 12 seedling-related parameters of rice.

SOV	Genotype (G)	Treatment (T)	G × T	Error	Total	Heritability (H^2^)
DF	23	1	23	92	143	
SL	15.15 ***	80.27 ***	4.72 *	2.58		72.68
RL	7.77 ^ns^	196.88 ***	5.11 ^ns^	6.64		14.42
TPL	27.22 **	528.6 ***	14.1 ^ns^	13.07		50.09
SFW	0.0022 ***	0.2007 ***	0.0005 ^ns^	0.008		65.60
RFW	0.0012 **	0.2097 ***	0.0003 ^ns^	0.006		52.69
SDW	0.0002 ***	0.0126 ***	0.0001 ***	0.000		72.39
RDW	0.7205 ^ns^	0.0367 ^ns^	0.7828 ^ns^	0.7519		0.001
SNC	4.05 ***	665.64 ***	6.6 ***	0.0864		37.57
SKC	3.65 ***	108.16 ***	1.44 ***	0.1137		71.01
RNC	0.211 ***	15.34 ***	0.224 ***	0.0067		47.68
RKC	0.424 ***	5.46 ***	0.455 ***	0.0068		47.85
NKR	0.23 ***	38.74 ***	0.29 ***	0.0087		43.69

*** Highly significant (*p* < 0.001); ** Highly significant (*p* < 0.01); * significant (*p* < 0.05); ns = non-significant. SOV = source of variation; DF = degree of freedom; G × T = Genotype (G) × Treatment (T) interaction; SL = shoot length (cm); RL = root length (cm); TPL = total plant length (cm); SFW = shoot fresh weight (g); RFW = root fresh weight (g); SDW = shoot dry weight (g); RDW = root dry weight (g); SNC = shoot sodium (Na^+^) concentration; SKC = shoot potassium (K^+^) concentration; RNC = root Na^+^ concentration; RKC = root K^+^ concentration; NKR = whole plant sodium/potassium (Na^+^/K^+^) ratio.

**Table 2 plants-11-01461-t002:** Performance of 24 rice genotypes under 200 mM salt stress condition in glasshouse.

Trait	Salt-Tolerant Genotypes Names and Mean Values	Salt-Susceptible Genotypes Names and Mean Values
SL	S13 (29.91), S1 (28.52), and S22 (27.14)	S3 (20.34), S11(22.92), and S8 (22.97)
RL	S5 (21.16), S3 (20.83), and S20 (20.41)	S8 (14.81), S24 (15.7), and S9 (16.51)
TPL	S4 (49.35), S5 (48.33), and S3 (47.6)	S24 (36.86), S9 (37.73), and S19 (39.83)
SFW	S4 (0.159), S5 (0.142), and S3 (0.141)	S24 (0.080), S9 (0.088), and S19 (0.094)
RFW	S12 (0.124), S24 (0.122), and S8 (0.113)	S17 (0.064), S7 (0.065), and S21 (0.066)
SDW	S5 (0.048), S2 (0.047), and S17 (0.045)	S9 (0.026), S19 (0.028), and S21 (0.029)
RDW	S18 (2.23), S10 (0.022), and S24 (0.020)	S23 (0.012), S3 (0.012), and S1 (0.013)
SNC	S21 (9.13), S5 (8.83), and S18 (8.63)	S15 (2.93), S22 (3.53), and S11 (3.6)
SKC	S24 (6.06), S15 (5.93), and S11 (5.46)	S20 (2.43), S7 (2.60), and S12 (2.63)
RNC	S17 (1.70), S22 (1.67), and S14 (1.62)	S24 (0.85), S12 (0.85), and S16 (0.92)
RKC	S10 (2.98), S1 (1.97), and S6 (1.91)	S17 (0.85), S23 (0.95), and S22 (1.15)
NKR	S15 (2.34), S10 (2.15), and S2 (2.12)	S3 (0.76), S16 (0.83), and S4 (1.01)

SL = shoot length (cm); RL = root length (cm); TPL = total plant length (cm); SFW = shoot fresh weight (g); RFW = root fresh weight (g); SDW = shoot dry weight (g); RDW = root dry weight (g); SNC = shoot sodium (Na^+^) concentration; SKC = shoot potassium (K^+^) concentration; RNC = root Na^+^ concentration; RKC = root K^+^ concentration; NKR = whole plant sodium/potassium (Na^+^/K^+^) ratio.

**Table 3 plants-11-01461-t003:** Mean squares values from Analysis of variance (ANOVA) for 7 maturity related parameters of rice.

SOV	Genotype (G)	Treatment (T)	G × T	Error	Total	Heritability
DF	23	1	23	92	143	
PH	282.9 ***	31.0 ^ns^	14.0 ^ns^	13.63		93.21
TN	28.3 **	1614.7 ***	10.19 ^ns^	12.10		57.23
GY	391.56 ***	104959.36 ***	151.45 ^ns^	151.40		61.32
SY	861.98 **	12886 ***	439 ^ns^	339.18		54.35
TGW	39.21 ***	205.76 ***	3.73 ^ns^	2.01		90.89
GL	4.30 ***	0.166 ^ns^	0.155 ^ns^	0.10		96.43
HI	0.0056 ***	0.8445 ***	0.0045 ***	0.001		48.10

*** Highly significant (*p* < 0.001); ** Highly significant (*p* < 0.01); ns = non-significant. SOV = source of variation; DF = degree of freedom; G × T = Genotype (G) × Treatment (T) interaction; PH = plant height (cm); TN = tillers number; GY = grain yield/plant (g); SY = straw yield (g); TGW = 1000-grain weight (g); GL = grain length (mm); HI = harvest index (%).

**Table 4 plants-11-01461-t004:** Performance of 24 GSR lines under field conditions.

Trait	Salt-Tolerant Genotypes Names and Mean Values	Salt-Susceptible Genotypes Names and Mean Values
PH	S3 (117.53), S23 (114), and S24 (113.8)	S18 (88.33), S8 (92.13), and S5 (93.33)
TN	S9 (22.2), S19 (22), and S24 (21.8)	S4 (14.33), S2 (14.73), and S6 (15)
GY	S9 (51.46), S8 (48.8), and S22 (42.66)	S1 (23.93), S6 (24.83), and S24 (27.2)
SY	S1 (125.6), S20 (115.06), and S23 (108.93)	S8 (43.6), S17 (64.26), and S19 (70.66)
TGW	S1 (30.62), S20 (24.63), and S19 (24.12)	S4 (17.43), S6 (17.63), and S12 (17.99)
GL	S1 (12.18), S12 (11.14), and S15 (10.05)	S2 (7.81), S10 (8.49), and S23 (8.81)
HI	S12 (0.41), S14 (0.37), and S9 (0.36)	S2 (0.15), S5 (0.20), and S1 (0.22)
SSI	S19 (1.16), S18 (0.94), and S23 (0.94)	S16 (0.743), S17 (0.746), and S2 (0.759)

PH = plant height (cm); TN = tillers number; GY = grain yield/plant (g); SY = straw yield (g); TGW = 1000-grain weight (g); GL = grain length (mm); HI = harvest index (%); SSI = stress susceptibility index.

## Data Availability

All the relevant data are within the paper and its Appendix A.

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
