# Peer review of "Evaluation of Green Super Rice Lines for Agronomic and Physiological Traits under Salinity Stress"

_plants, 2022, doi:10.3390/plants11111461_

Round 1

Reviewer 1 Report

The manuscript entitled „Evaluation of Green Super Rice Lines for Agronomic and Physiological Traits under Salinity Stress “presents a comprehensive study on salt tolerant variants of rice and molecular mechanisms underlying salt tolerance. The study encompasses experiments in glasshouse, as well as in the field, and is supplemented by expression analysis of genes potentially involved in salt tolerance. The introduction brings relevant information and materials and methods are described in sufficient detail. The study is relevant, within the scope of Plants and will be of interest to its readers. Therefore, I would recommend the manuscript for publication, but with following changes to make the manuscript more readable and reduce the unnecessary repeating within the text: All data should be primarily presented in a form of tables or figures. It is unnecessary to state the data in detail again in the text of the manuscript (e.g. page 6, page 11). The relevance and importance of data should be discussed in the text instead of just repeating the numbers. It would be recommendable to combine results and discussion sections to avoid unnecessary repeating. The discussion should not be another iteration of data, but rather, it should be expanded with a more comprehensive reflection of result obtained in this study within the context of other published studies on salt tolerance in rice.

Author Response

Response to Reviewer 1 Comments

The manuscript entitled “Evaluation of Green Super Rice Lines for Agronomic and Physiological Traits under Salinity Stress” presents a comprehensive study on salt tolerant variants of rice and molecular mechanisms underlying salt tolerance. The study encompasses experiments in glasshouse, as well as in the field, and is supplemented by expression analysis of genes potentially involved in salt tolerance. The introduction brings relevant information and materials and methods are described in sufficient detail. The study is relevant, within the scope of Plants and will be of interest to its readers. Therefore, I would recommend the manuscript for publication, but with following changes to make the manuscript more readable and reduce the unnecessary repeating within the text:

Dear Reviewer, Thank you for your interest in our manuscript. We also thank full for careful reading of our manuscript and your helpful comments and suggestions/questions. We have carefully revised this manuscript according to yours comments. A point-by-point reply to these comments are follows:

All data should be primarily presented in a form of tables or figures. It is unnecessary to state the data in detail again in the text of the manuscript (e.g. page 6, page 11). The relevance and importance of data should be discussed in the text instead of just repeating the numbers.

Response: Thank you for kind suggestion. As per suggestion unnecessary text is removed. Please see the revised version of the manuscript. 

It would be recommendable to combine results and discussion sections to avoid unnecessary repeating. The discussion should not be another iteration of data, but rather, it should be expanded with a more comprehensive reflection of result obtained in this study within the context of other published studies on salt tolerance in rice.

Response: Thank you for kind suggestion. But separate discussion is the requirement of the Journal (Plants). The unnecessary text is removed from the manuscript and we believed that now this revised version will be acceptable. 

Reviewer 2 Report

Amanat and coworkers show a detailed analysis of phenotypical data of different rice lines when grown under greenhouse and field conditions, and treated with control and salt solutions.  Overall, manuscript is well written, results are clearly exposed and coherent. Previous works using similar data analysis strategy have been published, what enforces the methodology used by Amanat and co-workers. Moreover, although some discrepancies, both greenhouse and field data are in agreement with the data obtained. Finally, different salt-response behaviors were found among the tested lines, and gene expression patterns support the observed phenotypes, and point to the resistance mechanisms that may be involved. In my opinion, the manuscript is suitable for publication in Plants, although some clarifications would make the whole manuscript more understandable for a general audience. Next, I will give the authors some recommendations for its consideration for final publication of the manuscript.

Material and methods section can be improved/completed.

Line 122- please, specify how plants were salt-treated. Were they watered with salt solutions?

Line 144- the term “normal field” is inaccurate. I would rather name it “control field conditions”. Same in line 157

Lines 126 and 127. The way dry weigh was estimated is not described. Did authors oven-dried samples?

Indicate how heritability was estimated in the M&M section.

Supplementary tables 4-6 are not found.

When results are described, a brief sentence explaining the main aim of the experiment, and a final conclusion would help to the understanding of the whole section.

Paragraph “3.2. Mean Variability of Seedlings Parameters” of Results section is obtuse to read. In my opinion, authors should make clear what the point of the analysis they did is. To me it is clear that there are two different analyses of these data. First, what are the most resistant and susceptible ecotypes for each character. And second, how variable are different phenotypes analyzed among genotypes under both control and saline conditions. These facts must be clear in this paragraph in order to be able to interpret the obtained data. Since data is quiet bast, few brief conclusions must be obtained and included in this paragraph.

Figure 1. Quality is not great. Little letters are not legible at all. Figure 1 foot notes on lines 284-285 must be more descriptive and accurate.

Lines 295-302. Regarding correlation analysis, results are not clearly exposed, and the main objective of this analysis is not clear. It would be recommended that authors clarify what they look for with this analysis, and a general brief conclusion of this analysis. In my opinion, phenotypes with a low correlation under control conditions and a high correlation under saline conditions, would be a good target for their purpose. Is this what authors considered? If so, please clarify it along this results paragraph. Therefore, it would be appropriated to make clear 1) what phenotypes are affected by salt treatment in all genotypes, and 2) according to correlation data, what phenotypes are more suitable to be used for assessing salt tolerant genotypes. Same for data collected from field grown plants.

PCA analysis from greenhouse and field collected data do not include same genotypes within the same groups (for instance, genotypes 13, 23, 1 and 19 in greenhouse conditions are grouped within the same group, whereas 13 and 23 are different than 1 and 19 for field analysis. It seems a discrepancy to me, although authors did not mentioned it. Can you discuss this fact?

Transcriptomic analysis shows different behavior of salt responsive genes. Some are downregulated only in susceptible lines, some others are activated only in resistant plants. Some other are activated in both. These results are interesting, and a better discussion would increase the interest of the manuscript (i.e. is the function of these genes reported? Why do authors observe this differential behavior?)

Some format comments:

Line 39 on abstract: “phosphoribosylaminomethylideneamino“ must be separated words. Is there any other shorter way to name this enzyme? It would be easier to read the abstract.

Line 83: the osmotic pressure of in vacuoles

Line 92. Indicate what GSR stands for.

Authors use the word “local check” to refer “local varieties” or “control varieties” I would suggest to use a more appropriated word instead of check

Line 136- parenthesis for the reference are not needed. This applies for some further references along the manuscript.

Line 123- phenotypes instead of genotypes?

Some extra-points are found along the text. i.e. lines 156 and 161.

Line 215-218- Although the overall meaning of this statement is clear, I would encourage authors to check the grammar on this sentence.

Lines 290-291 Check grammar

Morpho-physiological and Morphophysiological appears indistinctly along the text.

In my opinion, identification of different blocks of results (A, B and C) is not needed along the manuscript (i.e. lines 440, 333, etc)

Line 508-check grammar

Line 518. Avoid using showed two times in a row.

Line 541- “is present in plants roots”, use another verb to refer activity of a gene (i.e. expressed).

Author Response

Response to Reviewer 2 Comments

Amanat and coworkers show a detailed analysis of phenotypical data of different rice lines when grown under greenhouse and field conditions, and treated with control and salt solutions.  Overall, manuscript is well written, results are clearly exposed and coherent. Previous works using similar data analysis strategy have been published, what enforces the methodology used by Amanat and co-workers. Moreover, although some discrepancies, both greenhouse and field data are in agreement with the data obtained. Finally, different salt-response behaviors were found among the tested lines, and gene expression patterns support the observed phenotypes, and point to the resistance mechanisms that may be involved. In my opinion, the manuscript is suitable for publication in Plants, although some clarifications would make the whole manuscript more understandable for a general audience. Next, I will give the authors some recommendations for its consideration for final publication of the manuscript.

Dear Reviewer, Thank you for your time and for careful reading of our manuscript. We have carefully revised this manuscript according to yours comments. A point-by-point reply to these comments are follows:

Material and methods section can be improved/completed.

Line 122- please, specify how plants were salt-treated. Were they watered with salt solutions?

Response: Thank you for pointing this. Plants were treated with salt solution. Please see the lines 123-125.

Line 144- the term “normal field” is inaccurate. I would rather name it “control field conditions”. Same in line 157

Response: Thank you for kind suggestion. Suggestion is incorporated. Please see the lines 149 and 164.

Lines 126 and 127. The way dry weigh was estimated is not described. Did authors oven-dried samples?

Response: Thank you for pointing this. Seedlings were kept at 37 °C in oven for three days and dried weights were recorded by using electronics balance. Please see the lines 131-132.

Indicate how heritability was estimated in the M&M section.

Response: Thank you for pointing this. Heritability was calculated by using “heritability” package of R. Please see the lines 215-16.

Supplementary tables 4-5 are not found.

Response: Thank you for pointing this. Supplementary tables S1-S5 are provided. Please see the supplementary file.

When results are described, a brief sentence explaining the main aim of the experiment, and a final conclusion would help to the understanding of the whole section.

Paragraph “3.2. Mean Variability of Seedlings Parameters” of Results section is obtuse to read. In my opinion, authors should make clear what the point of the analysis they did is. To me it is clear that there are two different analyses of these data. First, what are the most resistant and susceptible ecotypes for each character. And second, how variable are different phenotypes analyzed among genotypes under both control and saline conditions. These facts must be clear in this paragraph in order to be able to interpret the obtained data. Since data is quiet bast, few brief conclusions must be obtained and included in this paragraph.

Response: Thank you for kind suggestion. Suggestion is incorporated. Please see the paragraph 3.2

Figure 1. Quality is not great. Little letters are not legible at all. Figure 1-foot notes on lines 284-285 must be more descriptive and accurate.

Response: Thank you for pointing this. Figures 1 and 4 are updated and description is added according to the suggestion. Please see the Figures 1 and 4.

Lines 295-302. Regarding correlation analysis, results are not clearly exposed, and the main objective of this analysis is not clear. It would be recommended that authors clarify what they look for with this analysis, and a general brief conclusion of this analysis. In my opinion, phenotypes with a low correlation under control conditions and a high correlation under saline conditions, would be a good target for their purpose. Is this what authors considered? If so, please clarify it along this results paragraph. Therefore, it would be appropriated to make clear 1) what phenotypes are affected by salt treatment in all genotypes, and 2) according to correlation data, what phenotypes are more suitable to be used for assessing salt tolerant genotypes. Same for data collected from field grown plants.

Response: Thank you for pointing this. Suggestion is incorporated, please see the headings 3.4 and 3.7.

PCA analysis from greenhouse and field collected data do not include same genotypes within the same groups (for instance, genotypes 13, 23, 1 and 19 in greenhouse conditions are grouped within the same group, whereas 13 and 23 are different than 1 and 19 for field analysis. It seems a discrepancy to me, although authors did not mention it. Can you discuss this fact?

Response: Thank you for pointing this. In the PCA analysis, genotypes scattered into different groups on the bases of overall performance/different traits. In the glasshouse there was a control conditions and environmental influence was minimum but in the field this was vice versa. The observed different grouping might be due to environmental effects but overall same kind of trend was observed in glasshouse and field experiments. On the bases of results from both experiments common tolerant genotypes (S1, S19, S20, S23, and S13) and susceptible genotypes (S24 and S3) were reported.

Transcriptomic analysis shows different behavior of salt responsive genes. Some are downregulated only in susceptible lines; some others are activated only in resistant plants. Some other are activated in both. These results are interesting, and a better discussion would increase the interest of the manuscript (i.e., is the function of these genes reported? Why do authors observe this differential behavior?)

Response: Thank you for pointing this. To understand the mechanism of salinity tolerance in rice, previously well known salt tolerant genes LOC_Os01g20160 (OsSKC1/OsHKT8), LOC_Os03g37930 (OsHAK21/qSE3), and LOC_Os02g52780 (OsbZIP23) were selected. It is reported that these genes upregulated in tolerant genotypes under salt stress. In this study, five novel genes from the previously published RNA-seq data were also selected to check their role in salinity tolerance. We observed that 4 out of 5 selected genes were upregulated in salt tolerant genotypes as compared to susceptible, and their expression was variable this might be due to the difference of genotypes or different tolerance levels of the genotypes. To confirm this further future experiments will be required.

Some format comments:

Line 39 on abstract: “phosphoribosyl amino methylidene amino” must be separated words. Is there any other shorter way to name this enzyme? It would be easier to read the abstract.

Response: Thank you for pointing this. Corrected as per suggestion. Please see the line 42.

Line 83: the osmotic pressure of in vacuoles

Response: Thank you for pointing this. Corrected as per suggestion. Please see the line 83.

Line 92. Indicate what GSR stands for.

Response: Thank you for kind suggestion. Full form is provided as per suggestion. Please see the line 92.

Authors use the word “local check” to refer “local varieties” or “control varieties” I would suggest to use a more appropriated word instead of check

Response: Thank you for nice suggestion. We used “local varieties” instead of “local checks”.

Line 136- parenthesis for the reference are not needed. This applies for some further references along the manuscript.

Response: Thank you for nice suggestion. Corrected as per suggestion. Please see the line 141.

Some extra-points are found along the text. i.e., lines 156 and 161.

Response: Thank you for nice suggestion. Corrected as per suggestion.

Line 215-218- Although the overall meaning of this statement is clear, I would encourage authors to check the grammar on this sentence.

Response: Thank you for nice suggestion. Corrected as per suggestion. Please see the lines 230-233.

Lines 290-291 Check grammar

Response: Thank you for nice suggestion. Corrected as per suggestion. Please see the lines 353-356.

Morpho-physiological and Morphophysiological appears indistinctly along the text.

Response: Thank you for nice suggestion. We used Morph-physiological and corrected as per suggestion.

In my opinion, identification of different blocks of results (A, B and C) is not needed along the manuscript (i.e., lines 440, 333, etc.)

Response: Thank you for nice suggestion. Corrected as per suggestion. Please see the revised manuscript.

Line 508-check grammar

Response: Thank you for nice suggestion. Corrected as per suggestion.

Line 518. Avoid using showed two times in a row.

Response: Thank you for nice suggestion. Corrected as per suggestion. Please see the line 628.

Line 541- “is present in plants roots”, use another verb to refer activity of a gene (i.e., expressed).

Response: Thank you for nice suggestion. Corrected as per suggestion. Please see the line 649.
